# High Probability of Long Diagnostic Delay in Coronavirus Disease 2019 Cases with Unknown Transmission Route in Japan

**DOI:** 10.3390/ijerph17228655

**Published:** 2020-11-21

**Authors:** Tsuyoshi Ogata, Hideo Tanaka

**Affiliations:** 1Tsuchiura Public Health Center of Ibaraki Prefectural Government, Tsuchiura 300-0812, Japan; 2Fujiidera Public Health Center of Osaka Prefectural Government, Fujiidera 583-0024, Japan; TanakaH61@mbox.pref.osaka.lg.jp

**Keywords:** COVID-19, diagnostic delay, transmission route, unknown exposure, Japan

## Abstract

Long diagnostic delays (LDDs) in patients with coronavirus disease 2019 (COVID-19) might decrease the effectiveness of patient isolation in reducing subsequent transmission. We assumed that direction of government considerably increased probability of LDD among COVID-19 cases with unknown exposure in Japan. This study aimed to investigate association of route of case detection and proportion of LDD of COVID-19 in Japan. We included confirmed COVID-19 patients with symptom onset between the ninth and eleventh week in 2020, in 6 prefectures of Japan. LDD was defined as the duration between COVID-19 symptom onset and confirmation ≥6 days. We used multivariable logistic regression analyses to elucidate factors associated with LDD. The mean diagnostic delay for 364 cases was 6.3 days. Proportion of LDD was 38% for cases with known exposure, and 65% for cases with unknown exposure. The probability of LDD in cases with unknown exposure was significantly higher than that for known exposure cases (adjusted odds ratio: 2.38, 95% confidence interval: 1.354–4.21). Early PCR test after symptom onset, strengthening of PCR test capacity, and investigations to study impact of high proportion of LDD in cases without known exposure might be necessary.

## 1. Introduction

An outbreak of novel coronavirus disease 2019 (COVID-19) began in December 2019 in China and spread to other countries including Japan, and the World Health Organization declared it a public health emergency of international concern [1].

A study using mathematical model showed that the delay between symptom onset and isolation had a major role for controlling COVID-19 outbreak, and that probability of achieving the outbreak control fell when isolation delay became long, 8.1 days, compared to short delay, 3.4 days [2]. In Taiwan, the secondary attack rate was low among those initially exposed to infected cases after 6 days of symptom onset [3]. Long diagnostic delay (LDD) such as ≥6 days might largely decrease the effectiveness of intervention involving patients’ isolation to reduce the risk of subsequent transmissions.

In Japan, public health centers eagerly implemented contact tracing under direction of the “Cluster response task force” of the National Government in the first wave and contacts of known exposure cases underwent polymerase chain reaction (PCR) tests relatively fast. As for patients without known exposure to COVID-19, on the other hand, the National Government directed with an educational leaflet that the mild patients with concerns of contracting COCID-19 infection should consult a public health center or visit a physician basically only after waiting 4 days from symptom onset [4,5,6,7,8]. The “criterion of waiting 4 days” might increase proportion of LDD, and subsequently the number of local patients.

We assumed that proportion of LDD was considerably large among COVID-19 cases with unknown exposure compared to cases with known exposure in Japan. The aim of this study was to elucidate association of route of case detection and proportion of LDD of COVID-19 in Japan. We thought the result was significant for evaluating the early response to COVID-19 as well as improving response to further COVID-19 waves or a pandemic in the future.

## 2. Materials and Methods

### 2.1. Study Design

The study used a cross-sectional study design.

### 2.2. Setting and Subject

In Japan, the first COVID-19 patient, who had previously visited Wuhan, China, was identified on January 15, 2020 [4]. Many of the COVID-19 patients at the earliest stage in Japan had been exposed to the virus in China, or on a cruise ship (Diamond Princess) and were quarantined off the Yokohama port in Japan [9,10]. However, an increase in the number of COVID-19 patients who had not visited China nor were present on the cruise ship was reported in Japan in the latter half of February 2020. The number of cases rapidly increased by March 2020 [4].

The Ministry of Health, Labor and Welfare regulated subjects for the SARS-CoV-2 PCR test on February 17 in the eighth week of 2020. Such patients included those suspected of infection through an unknown exposure by a physician, a close contact of a COVID-19 patient, or one who had migrated from foreign countries with the epidemic [4].

Although the Governor of Hokkaido declared a situation of emergency and requested people to stay at home on February 28 (the ninth week), 2020 as a response against the increased number of COVID-19 patients [11], no such interventions, except school closure in the whole nation, were implemented until the 11th week in other prefectures of Japan. However, as the number of COVID-19 cases increased in the 12th week, the Governor of Osaka Prefecture requested the populace not to commute between Osaka Prefecture and Hyogo Prefecture (12th week). The Governor of Tokyo also requested physical distancing on March 25 (13th week), one day after the joint statement rescheduling the 2020 Olympiad in Tokyo [12]. Because of these responses, situations regarding diagnostic delay of COVID-19 might have changed after the 12th week in some prefectures. Therefore, we selected the symptomatic COVID-19 incident cases with dates of onset of up to the 11th week (March 15) in 2020. Patients with symptom onset in the 11th week were reported in the 12th week at most.

Japan consists of 47 prefectures. In this study, we included those prefectures with >30 COVID-19 patients reported as of March 22, 2020 (the end of the 12th week). Therefore, eight prefectures, Hokkaido, Saitama, Chiba, Tokyo, Kanagawa, Aichi, Osaka, and Hyogo, were eligible. However, we could not retrieve the data on the route of exposure of patients in Osaka and Tokyo. Finally, 6 prefectures (population), Hokkaido (5.2 million), Saitama (7.3 million), Chiba (6.3 million), Kanagawa (9.2 million), Aichi (7.6 million), and Hyogo (5.5 million), were eligible.

Although in 2 out of the 6 prefectures, the number of cases with symptom onset at the end of the eighth week was less than 5, the number of cases with symptom onset at the end of the ninth week in all the 6 prefectures were not less than 5. Therefore, we selected symptomatic COVID-19 cases with a date of onset in and after the ninth week (beginning on February 24, 2020), in each prefecture. Finally, we included study subjects whose date of onset was between the 9th and the 11th week, in 2020. This study included COVID-19 patients in the 6 prefectures whose symptoms onset was dated between February 24 and March 15 (the 9th and 11th week), 2020, and SARS-CoV-2 positive patients as confirmed by a PCR test. We excluded cases who were asymptomatic and those whose symptom onset dates were missing.

### 2.3. Data Collection and Variables

We only used retrieved publicly available anonymized data from 23 local governments’ official websites in the 6 prefectures (Table A1
Appendix A). The data included sex, age category, date of onset of subjective symptoms, date of confirmation of SARS-CoV-2 positivity by PCR test, and exposure route (route of case detection). Route of exposure was classified based on the information disclosed at the time of diagnosis into “known” (having had contact with an infected case or visited a place of an outbreak), “imported” (from foreign countries during the incubation period), and “unknown.” We censored the data after April 6, 2020 (the end of the 14th week) because data in 2 prefectures could not be retrieved in the 15th week.

### 2.4. LDD

The timeline in reporting an infectious disease generally includes the time from symptom onset to visiting a physician, time from visiting a physician to the laboratory confirmation, and time from the laboratory confirmation to making an official report [13,14]. During the study period, the Japanese physicians notified the relevant public health center (which belongs to a local municipal government) if they suspected a patient of having COVID-19. At the public health centers, clinical samples were collected from the patient and submitted to the prefectural institutes of public health for assay. If the sample was SARS-CoV-2 positive, the public health center instantly admitted the patient in a designated hospital or was isolated at home. Therefore, we defined diagnostic delay in this study as the duration between the date of symptom onset and date of confirmation of SARS-CoV-2 positivity by PCR test. If both the dates of symptom onset and confirmation occurred on the same day, we allotted the delay value as 0.5 days. We defined a case LDD if the duration of COVID-19 symptom onset and confirmation is ≥6 days in this study because such a delay might largely reduce the intervention (patient isolation) effect [3].

### 2.5. Statistical Analysis

To confirm that the obtained data represented the situation in each prefecture, we calculated the proportion of symptomatic cases with unknown onset date to all the SARS-CoV-2 positive cases, reported between March 2 (the 10th week) and March 22 (the 12th week) in each prefecture; since the incubation period is approximately a week.

We calculated the mean and 95% confidence interval (95% CI) of diagnostic delay, without adjusting for right truncation, by 1000 times bootstrapping and by fitting parametric distributions to diagnostic delay in a Bayesian framework.

We described the characteristics of subjects with COVID-19. We used the multivariable logistic regression model analyses to calculate the proportion of patients with LDD and the adjusted odds ratios with 95% CI. We compared them across the week of symptom onset, route of case detection, and 6 prefectures. We implemented similar analyses using ≥5 days and ≥7 days instead of ≥6 days, as the definition of LDD.

Statistical analyses were performed using R (version 3.6–2; The R Foundation for Statistical Computing, Vienna, Austria).

### 2.6. Ethical Concerns

The study used only publicly available anonymized data. The study did not use patients’ data from other sources. The study was approved by the Tsuchiura Public Health Center of Ibaraki Prefectural Government (protocol number: Tsuchi-Ho R20–01).

## 3. Results

The proportion of symptomatic cases that we could not acquire the data on the date of onset of COVID-19 reported between March 2 and March 22 (the 10th and 12th week) in each prefecture ranged from 0% in Saitama Prefecture to 8.5% in Osaka Prefecture. The onset of symptoms or diagnostic delay was obtained for most COVID-19 patients during the period, in each prefecture. We included 364 COVID-19 patients in the 6 prefectures whose symptom onset was dated between February 24 and March 15 (the 9th and 11th week), 2020.

The mean diagnostic delay of the 364 COVID-19 patients was 6.28 days (95% CI: 5.8–6.8) with a standard deviation (SD) of 4.57 days (95% CI: 4.1–5.0) by bootstrapping. The mean diagnostic delay was 6.28 days (95% Credible interval (CrI) 5.3–8.4) by fitting gamma distribution, 6.84 days (95% CrI: 5.8–8.1) by fitting Weibull distribution, and 6.19 days (95% CrI: 5.7–6.8) by fitting log-normal distribution with the lowest Akaike’s Information Criterion among three distributions.

The males were 190 (52%), and those aged ≥60 years were 196 (54%). The cases with known exposure, the cases with unknown exposure, and imported cases were 209 (57%), 118 (32%), and 37(10%), respectively. (Table 1).

The total proportion of LDD was 51%. Proportion of LDD was 38% for cases with known exposure, 65% for cases with unknown exposure, and 73% for imported cases. The adjusted odds ratios of LDD were 0.31 (95% CI: 0.170–0.58) and 0.17 (95% CI: 0.090–0.32) for patients with symptom onset in the 10th and 11th week, respectively, compared to the 9th week. The adjusted odds ratios of LDD were 2.38 (95% CI: 1.354–4.21) and 3.51 (95% CI: 1.418–8.75) for unknown and imported cases, respectively, compared to patients with known exposure. The adjusted odds ratios were significantly higher for patients in the other five prefectures than patients in Aichi (Table 2).

When we changed the definition of LDD from ≥6 days to ≥5 days or ≥7 days, the adjusted odds ratios of LDD were also significantly higher for cases with unknown exposure (4.18 [95% CI: 2.22–7.88] and 1.96 [95% CI: 1.12–3.44], respectively) and for imported cases (11.1 [95% CI: 2.89–43.0] and 2.77 [95% CI: 1.17–6.52], respectively), compared to cases with known exposure.

## 4. Discussion

The mean diagnostic delay between symptom onset and COVID-19 confirmation was approximately 6.3 days between the 9th and 11th week in Japan; it was longer than 6 days. The mean delay was shorter than 6 days in Shenzhen (4.6 days), Hong Kong (3.2 days), Korea (4.3 days), and Singapore (5.6 days) [15,16,17,18]. The figures could be influenced by censoring of date collection and definition of delay.

While proportion of LDD was 38% for cases with known exposure, it was 65% for cases with unknown exposure in Japan. The probability of LDD in unknown exposure was significantly higher than that for known exposure cases, with adjusted odds ratios of 2.38. In Shenzhen, the average isolation delay of cases detected through symptom-based surveillance was shorter than 6 days though it was longer than that of cases detected through contact-based surveillance [15].

We assume the large proportion of LDD for cases with unknown exposure in Japan decreased the effectiveness of patient isolation, and subsequently increased the number of local patients.

We think that the high proportion of LDD in cases without known exposure to COVID-19 was largely because the National Government directed the patients with the “criterion of waiting 4 days” before consulting at a public health center or visiting a physician [4,5,6,8]. Though we dealt 6 prefectures due to the limited data availability, National Government’s “criterion of waiting for 4 days” could delay the PCR test for unknown exposure cases broadly in Japan. We could not find any other major situation to change this finding in generalizing to other prefectures. Therefore, we think this finding can apply to the whole Japan.

We could further consider several circumstances behind “criterion of waiting 4 days” for PCR testing in Japan. First, the number of PCR tests per population was small compared to those of other countries in Japan [19]. It reflected relatively small capacity for implementing PCR test in Japan [7,8]. Japan Medical Association reported inappropriate cases for PCR test, on March 17th, which included refusal by public health center to implement PCR test in spite of request by primary care physician [20]. Second, clinical specialists on infectious diseases in Japan seemed to be reluctant to swiftly find COVID-19 cases. The Japanese Association for Infectious Diseases and Japanese Society for Infection Prevention and Control claimed that “PCR testing is basically not recommended for mild cases. If the disease tends to become more severe over time, consider performing the PCR method” [21]. The opinion was reflected in government policy through governmental advisory board [8]. Third, epidemiologists of “Cluster response task force” emphasized “targeted testing strategy” and early identification of clusters [4,8]. The policy of the National Government based on them. However, “targeted testing strategy” might be yet to be evaluated enough.

We think the results help in pandemic preparedness in further waves as well as future pandemics. The authors propose following recommendations: First, for preventing further spread of transmission, PCR test should be implemented earlier from symptom onset for patients even if they do not have any epidemiological link with COVID-19. Second, capacity to implement PCR test should be strengthened [8]. Japan considerably depends on SARS-CoV-2 antigen test, instead of PCR test, for increasing capacity of SARS-CoV-2 test in October, 2020. However, developing novel antigen test for emerging infectious disease would take time. Therefore, basic capacity for implementing PCR test should also be increased for preparing for future pandemic in Japan. Third, further risk communication should be made between public health sector and clinical specialists of infectious disease to mutually understand adequate timing of test for preventing transmission as well as clinical management. Fourth, further investigations are necessary to study impact of high proportion of LDD in cases without known exposure on the subsequent increase in the number of COVID-19.

This study had several limitations. First, this study used cross-sectional design; thus, the results did not prove any causal relationships. Second, the data were collected through websites only and could not be confirmed for their authenticity. Third, differences in management and disclosure of relevant data might exist among the prefectures that influenced the results. Finally, due to the limited number of PCR tests, the data did not necessarily capture all the cases.

## 5. Conclusions

The probability of LDD for patients without known exposure was 65% and significantly higher than that for known exposure cases in Japan. Early PCR test from symptom onset for them and strengthening of capacity to implement PCR test are recommended. Further investigations are necessary to study impact of high proportion of LDD in cases without known exposure on the subsequent increase in the number of COVID-19.

## Figures and Tables

**Table 1 ijerph-17-08655-t001:** Characteristics of symptomatic COVID-19 cases in Study 1.

Factors		Cases	
		Number	%
		364	100.0
Sex	Male	190	52.2
	Female	174	47.8
Age category (years)	0–29	31	8.5
	30–59	137	37.6
	60-	196	53.8
Week of symptom onset	9th	113	31.0
	10th	130	35.7
	11th	121	33.2
Exposure	Known	209	57.4
	Unknown	118	32.4
	Imported	37	10.2
Prefecture	Aichi	102	28.0
	Hokkaido	84	23.1
	Hyogo	73	20.1
	Saitama	39	10.7
	Kanagawa	43	11.8
	Chiba	23	6.3

**Table 2 ijerph-17-08655-t002:** Factors associated with long diagnostic delay (≥6 days) in the symptomatic COVID-19 cases observed between 9th and 11th week, 2020, in Japan.

Factors		Reporting Delay			
		≥6 days	≤5 days	Univariate analysis	Multivariate analysis *
		(%)	(%)	Odds ratio (95% confidence interval)	Odds ratio (95% confidence interval)
N		184 (51%)	180 (49%)		
Sex	Male	94 (49%)	96 (51%)	1	1
	Female	90 (52%)	84 (48%)	1.09 (0.731–1.65)	1.58 (0.942–2.66)
Age (years)	0–29	19 (61%)	12 (39%)	1	1
	30–59	72 (53%)	65 (47%)	0.69 (0.321–1.55)	0.98 (0.392–2.51)
	60-	93 (47%)	103 (53%)	0.57 (0.261–1.24)	1.14 (0.452–2.87)
Week †	9th	81 (72%)	32 (28%)	1	1
	10th	59 (45%)	71 (55%)	0.33 (0.190–0.56)	0.31 (0.17–0.58)
	11th	44 (36%)	77 (64%)	0.23 (0.130–0.39)	0.17 (0.09–0.32)
Exposure	Known	80 (38%)	129 (62%)	1	1
	Unknown	77 (65%)	41 (35%)	3.03 (1.894–4.15)	2.38 (1.354–4.21)
	Imported	27 (73%)	10 (27%)	4.35 (2.009–9.47)	3.51 (1.418–8.75)
Prefecture	Aichi	18 (18%)	84 (82%)	1	1
	Hokkaido	47 (56%)	37 (44%)	5.93 (3.041–11.5)	4.53 (2.19–9.39)
	Hyogo	45 (62%)	28 (38%)	7.50 (3.751–15.0)	7.66 (3.61–16.3)
	Saitama	24 (62%)	15 (38%)	7.47 (3.281–17.0)	7.43 (2.86–19.3)
	Chiba	17 (74%)	6 (26%)	13.22 (4.583–38.2)	11.3 (3.543–6.1)
	Kanagawa	33 (77%)	10 (23%)	15.40 (6.443–36.8)	13.67 (5.17–36.1)

* All the factors listed above were included as independent variables in the logistic regression analysis; † Week of the date of onset.

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
