# Peer review of "High Probability of Long Diagnostic Delay in Coronavirus Disease 2019 Cases with Unknown Transmission Route in Japan"

_ijerph, 2020, doi:10.3390/ijerph17228655_

Round 1

Reviewer 1 Report

Even with some limitations in data availability and concerns about ecological fallacy as mentioned already by author(s), this study was conducted very systematically. There are some minor points requiring the authors’ attention.

  1. There are Study 1 and Study 2 in this paper but the relationships between these two studies and the roles of each study should be more clarified. And, what research question could be answered by each study needs to be elaborated on further.
  2. This study deals with only 6 prefectures due to the limited data availability but it needs to provide even a brief explanation about the generalizability of the findings to the other prefectures in Japan.
  3. What kinds of policies for preventing the spread of COVID-19 were adopted in the six prefectures should be mentioned even briefly.

Reviewer 2 Report

This study aims to investigate several factors that could be associated with long diagnostic delays (LDDs) in patients with coronavirus disease 2019 (COVID-19); moreover, it tries to analyze the correlation between LDD and subsequent doubling time.

The major concerns about this paper are related to the methods used to perform the study 2. 

The introduction section is too long. Under my opinion, it should be shortened, erasing the unnecessary parts. Particularly, it should be functional to the study's aims. Moreover, the study's aims should be better clarified. Please, check this section.

The material and methods indication described two studies. In Study 1, the authors reported that they have retrieved the data (sex, age category, date of onset of subjective symptoms, date of confirmation of SARS-CoV-2 positivity by PCR test, and exposure route) from the websites. Moreover, they described the statistical analysis performed.
In study 2, the authors calculated the doubling time (estimated by dividing the natural logarithm of 2 by the growth rate), reporting the test used for the analysis of data.

Based on the results of Study 1, the authors reported the data referred to 364 COVID-19: the mean diagnostic delay of the 364 COVID-19 patients was 6.28 days with a standard deviation (SD) of 4.57 days.
The results of Study 2 showed the geographic correlation between the proportion of long diagnostic delay and subsequent 14 days moving average doubling time of COVID-19 incidence (the 12-13th week) among the 6 Japanese prefectures by multivariate regression analysis.

The results section summarized the main findings of both studies.

Under my opinion, study 2 should be better described both in material and methods and in results. I believe that it is not clear the methods applied to obtain the described results. Particularly, these sections should be improved.
In the same manner, the discussion section about Study 2 should be improved. Moreover, the discussion section about Study 1 should be improved too. Particularly, the authors missed comparing their results to the international data. Please, improve this section.

Round 2

Reviewer 2 Report

Following the reviewers' suggestion, the authors have fixed all of my concerns. I believe that it could be published in the current form.
